# Quantized Disentangled Representations for Object-Centric Visual Tasks

## Abstract

Recently, the pre-quantization of image features into discrete latent variables has helped to achieve remarkable results in image modeling. In this paper, we propose a method to learn discrete latent variables applied to object-centric tasks. In our approach, each object is assigned a slot which is represented as a vector generated by sampling from non-overlapping sets of low-dimensional discrete variables. We empirically demonstrate that embeddings from the learned discrete latent spaces have the disentanglement property. The model is trained with a set prediction and object discovery as downstream tasks. It achieves the state-of-the-art results on the CLEVR dataset among a class of object-centric methods for set prediction task. We also demonstrate manipulation of individual objects in a scene with controllable image generation in the object discovery setting.

## 1 Introduction

The known problem of existing neural networks is that they cannot generalize at the human level (Lake et al. (2016); Greff et al. (2020)). It is assumed that the reason for this is the inability of current neural networks to dynamically and flexibly bind information distributed throughout it. This is called the *binding problem*. This problem affects the ability of neural networks 1) to construct meaningful representations of entities from unstructured sensory inputs; 2) to maintain obtained separation of information at a representation level; 3) to reuse these representations of entities for new inferences and predictions. One way to solve this problem is to constrain the neural network for learning disentangled object-centric representations of a scene (Burgess et al. (2019); Greff et al. (2019); Yang et al. (2020b)).

The disentangled object-centric representation may potentially improve generalization and explainability in many machine learning domains such as structured scene representation and scene generation (El-Nouby et al. (2019); Matsumori et al. (2021); Kulkarni et al. (2019)), reinforcement learning (Keramati et al. (2018); Watters et al. (2019a); Kulkarni et al. (2019); Berner et al. (2019); Sun et al. (2019)), planning (Migimatsu & Bohg (2020)), reasoning (Yang et al. (2020a)), and object-centric visual tasks (Groth et al. (2018); Yi et al. (2020); Singh et al. (2021)). However, recent research has focused either on object-centric or disentangled representation and has not paid enough attention to combining them. There are just several works that consider both objectives (Burgess et al. (2019); Greff et al. (2019); Li et al. (2020); Yang et al. (2020b)).

We propose a method that produces the disentangled representation of objects by quantization of the corresponding slot representation. We call it Vector Quantized Slot Attention (VQ-SA). VQ-SA obtains object slots in an unsupervised manner (Locatello et al. (2020)) and then perform quantization. The slot quantization involves two steps. At the first step, we initialize several discrete latent spaces each corresponding to the one of potential generative factors in the data. At the second step, we initialize each latent space with separate embeddings for potential values of a corresponding generative factor. This two-step quantization allows the model to assign a particular generative factor value to a particular latent embedding.

The proposed object-centric disentangled representation improves the results of the conventional model from Locatello et al. (2020) on object-centric visual tasks such as set prediction compared to light-weighted specialized models (Locatello et al. (2020); Zhang et al. (2019)). We demonstrate it through extensive experiments on the CLEVR dataset (Johnson et al. (2017)).

To measure degree of disentanglement the commonly used disentanglement metrics are BetaVAE score (Higgins et al. (2017a)), MIG (Chen et al. (2018)), DCI disentanglement (Eastwood & Williams (2018a)), SAP score (Kumar et al. (2018a)), and FactorVAE score (Kim & Mnih (2018)). These metrics are based on the assumption that disentanglement is achieved on a vector coordinate level, i.e. each coordinate corresponds to the generative factor. In our approach, generative factors are expressed by vectors, and separate coordinates are not interpretable. Thus, metrics listed above are not suitable and the problem of quantitative evaluation of disentanglement in the case of vector representation of generative factors remains an open question for future studies. Nevertheless, we propose DQCF-micro and DQCF-macro methods that qualitatively evaluate the disentanglement in the object discovery task. Original Slot Attention based model achieves remarkable results in the object discovery task, but our model allows not only to separate distributed features into object representations but also to separate distributed features of the objects themselves into representations of their properties.

We first give an overview of the proposed model VQ-SA (Section 2.1). Then, we provide a detailed explanation of the slot quantization approach (Section 2.3) we use to represent objects from an image. We conduct experiments on the CLEVR dataset (Johnson et al. (2017)) for the set prediction task (Section 3.1) and show that our model achieves the state-of-the-art results in some settings and performs comparably well in others. We also conduct experiments for the object discovery task (Section 3.2) and show quality results for the CLEVR dataset (Johnson et al. (2017)). We conduct ablation studies (Section 5) and provide results of modified versions of the proposed model to confirm our design choices. The learned discrete latent spaces possess the disentangled property. We qualitatively demonstrate this (Section 4) by analyzing set prediction results. Finally, we position our work relative to other approaches (Section 6) and discuss the obtained results, advantages, and limitations of our work (Section 7).

Our main contributions are follows:

- We propose a discrete representation (quantization) of object-centric embeddings (Section 2.3) that maps them to several latent spaces.

- The quantization produces disentangled representation (Section 4) there the disentanglement achieved on the level of latent embeddings rather than embedding coordinates.

- Learned discrete representations allow us to manipulate individual objects in a scene and generate scenes with objects with given attributes by manipulation in the latent space (Section 3.2).

- The proposed model VQ-SA achieves state-of-the-art results on the set prediction task on the CLEVR dataset (Section 3.1) among a class of object-centric methods.

- We propose DQCF-micro and DQCF-macro methods that qualitatively evaluate the disentanglement of the learned discrete variables, when they are represented by vectors rather by vector coordinates.

## 2 METHOD

### 2.1 OVERVIEW

To obtain valuable object representations, we should first discover objects in the image and then transform their representations into desired ones. We discover objects in an unsupervised manner with the use of a slot attention mechanism (Locatello et al. (2020)). The idea of slot representation is to map an input (image) to a set of latent variables (slots) instead of a single latent vector (Kingma & Welling (2014); Rezende et al. (2014)) such that each slot will describe a part of an input (Locatello et al. (2020); Engelcke et al. (2020; 2021)). We assign each object to a slot of a dimension $d_s$. Further, we transform each slot to a desired latent representation. We draw inspiration from the discrete latent representation proposed in van den Oord et al. (2017) and apply its modification to each slot. We use multiple latent spaces with small embedding dimension $d_l$ ($d_l < d_s$) and the small number of embeddings in each latent space instead of using a single discrete latent space to map slots. The small dimension of vectors in the latent spaces enables us to construct the resultant slot representation by concatenation. That could be seen as constructing a new vector of factors from given ones.

The main assumption behind this design choice is that each object is generated by a fixed number of generative factors. Thus, it is possible to represent each object (slot) as a combination of embeddings corresponding to values of generative factors. As most of the generative factors have a discrete nature, the proper choice of the generative distribution would be the categorical one. Particularly, we choose the number of categorical distributions equal to that of generative factors with the number of possible categories equal to that of values of a corresponding generative factor. The overall architecture of the proposed model VQ-SA is depicted in Fig. 1.

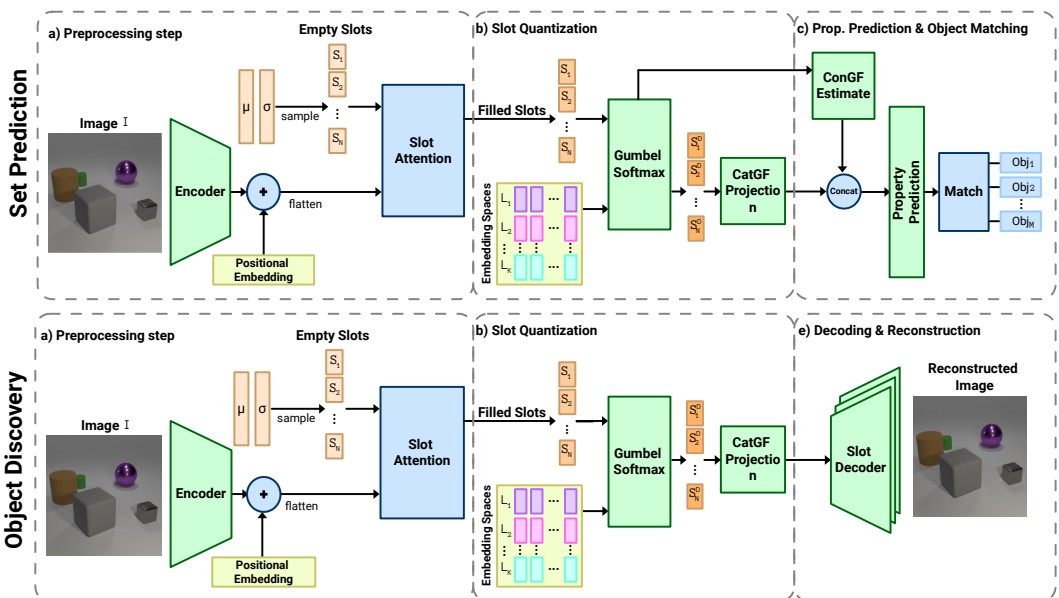

Figure 1: Overall architecture of the proposed VQ-SA mode. Top part corresponds to the set prediction task and bottom part corresponds to the object discovery task. We assign objects to slots in an unsupervised manner. The preprocessing and slot quantization part are the same in both tasks. To process categorical generative factors (CatGF), i.e., shape, color, size, material, we perform quantization by multiple discrete latent spaces with a Gumbel-Softmax trick. For continuous generative factors (ConGF), i.e., $x, y, z$ coordinates, we estimate their values (ConGF Estimate). In the case of the object discovery task, we treat all generative factors as categorical. For continuous factors, this corresponds to splitting it into bins. For the set prediction task, we process categorical and continuous generative factor separately. Then we concatenate both representations and predict object properties.

The VQ-SA pipeline for set prediction task consists of three stages. At the first preprocessing step (Fig. 1a), the image $I$ is flattened by Encoder, combined with positional embeddings, and objects are assigned to slots by an iterative slot attention mechanism. At the second stage (Fig. 1b), we perform separate processing of categorical generative factors (CatGF) and continuous generative factors (ConGF). We use slot quantization by multiple discrete latent spaces to represent categorical generative factors and estimate values of continuous generative factors with ConGF Estimate module. At the final, third stage, (Fig. 1c) we combine both types of generative factors and predict object properties. Then, we match the predicted objects with the ground truth using the Hungarian algorithm (Kuhn (1955)).

For the object discovery task, the pipeline differs at the third stages. We process continuous generative factors similar to generative factors (Fig. 1b) and use the same decoding strategy as in the original Slot Attention Locatello et al. (2020) (Fig. 1e).

## 2.2 SLOT REPRESENTATION

We follow the procedure proposed in Locatello et al. (2020) to represent an image $I$ as a set of slots $S_i, i = 1 \ldots N$, where the slots are vectors of dimension $d_s$. Each slot $S_i$ is initialized randomly $S_i = \mu + \sigma \cdot N(0, 1)$ and is used to assign an object $O_m, m = 1 \ldots M$ or a background from an

image in $I$ an iterative procedure. The main feature of a slot attention mechanism is that slots are competing with each other for assigning an object, i.e., attention coefficients are normalized over the slots. Following Locatello et al. (2020), we use dot-product attention (Luong et al. (2015)) as an attention mechanism for slots and aggregate the resultant slot representation by Gated Recurrent Unit (GRU) (Cho et al. (2014)) (Fig. 1).

## 2.3 SLOT QUANTIZATION

After assigning the objects to the slots, each slot $S_i$ is represented in discrete latent spaces $L_k, k = 1 \dots K$, $K$ is the number of latent spaces corresponding to the number of categorical generative factors ($K = 4$ in the case of the CLEVR dataset). Each latent space $L_k$ is initialized with embeddings $e_j^k, j = 1 \dots n_k$ with dimension $d_l$ ($d_l < d_s$). $n_k$ is the number of embeddings in the latent space $L_k$, is equal to the number of categories of a corresponding categorical generative factor. We linearly project a slot $S_i$ to a lower dimension $d_l$: $S_i' = M S_i$ ($S_i' \in \mathbb{R}^{d_l \times 1}$, $M \in \mathbb{R}^{d_l \times d_s}$, $S_i \in \mathbb{R}^{d_s \times 1}$). Then, we construct a new representation of a slot $S'$ in each discrete latent space $L_i$ by the Gumbel-Softmax trick (Jang et al. (2016); Maddison et al. (2016)). First, we calculate the similarity between the slot $S_i'$ and each embedding $e_j^k$ in latent space $L_k$ to get posterior distributions $q(e_j^k|S_i')$ by normalization with a Softmax function:

$$
\begin{aligned}
sim^k &= (S_i')^T L_k = (S_i')^T [e_1^k, e_2^k, \dots, e_{n_k}^k], \\
q(e_j^k|S_i') &= \frac{\exp(sim_j^k)}{\sum_j \exp(sim_j^k)}.
\end{aligned}
\tag{1}
$$

To get the continuous approximation $y^k$ of the one-hot-encoded representation of the discrete variable $e^k$ we use Gumbel-Softmax trick with a constant temperature parameter $t = 2$.

$$
y_j^k = \frac{\exp\left(\frac{(g_j + \log(q(e_j^k|S_i')))}{t}\right)}{\sum_j \exp\left(\frac{(g_j + \log(q(e_j^k|S_i')))}{t}\right)},
\tag{2}
$$

where $g$ denotes random samples from a Gumbel distribution.

The resultant representation $\hat{e}_i^k$ of slot $S_i'$ in the discrete latent space $L_k$ is a weighted sum of all embeddings $e_j^k \in L_k$ from $L_k$ with weights $y^k = [y_1^k, y_2^k, \dots, y_{n_k}^k]$:

$$
\hat{e}_i^k = (y^k)^T L_k = [y_1^k, y_2^k, \dots, y_{n_k}^k]^T [e_1^k, e_2^k, \dots, e_{n_k}^k]
\tag{3}
$$

Then, representations of $S_i'$ from all discrete latent spaces $L_k, k = 1 \dots K$ are concatenated:

$$
S_i^D = [\hat{e}_i^1, \dots, \hat{e}_i^K].
\tag{4}
$$

$S_i^D$ further is used as a quantized representation of the slot $S_i'$. Concatenation could be seen as a construction of a new vector representation $S_i^D$ from separate generative factors $\hat{e}_i^1, \dots, \hat{e}_i^K$.

## 2.4 COORDINATES REPRESENTATION

We use discrete latent spaces to represent generative factors with categorical nature. However, such encoding of continuous generative factors, e.g., coordinates, may lead to significant errors due to quantization. Also, there is no unique way to represent a continuous value by a discrete segment, as the length of a segment could be viewed as a hyperparameter.

In the set prediction task, we explicitly split generative factors into categorical (shape, color, size, material) and continuous ($x, y, z$ coordinates). We use slot quantization (Section 2.3) to represent categorical generative factors and obtain a quantized slot representation $S_i^D$. We use multilayer perceptron (MLP) with two layers to predict object coordinates.

In the object discovery task, we also process coordinates separately but in the manner similar to categorical generative factors (Section 2.3). The only difference is that we use a single latent space rather than multiple latent spaces.

## 2.5 Encouraging disentanglement

In order to aid disentanglement between discrete latent spaces, we add the following well-known term to the loss function:

$$-\sum_k \mathrm{KL}(q(L_k|S')\|p(L_k)) \tag{5}$$

$p(L_k)$ is a true prior uniform categorical distribution over discrete latent space $L_k$, $q(L_k|S')$ — a posterior categorical distribution over a discrete latent space predicted by a neural network. Using this loss term, we force posterior distributions over each latent space $q(L_k|S')$ to be independent and closer to prior distributions, which results in better disentanglement between spaces.

## 3 Experiments

In this section, we demonstrate the two advantages of using the additional vector quantization module. First, it improves performance on the set prediction task without drastically changing the overall model architecture and transforming it into a highly specialized set prediction model (Section 3.1). Second, unsupervised training for the object discovery task makes it possible to learn representations that could be used to edit individual objects in the image in an interpretable manner (Section 3.2).

For all tasks, we first trained the original Slot Attention (Locatello et al. (2020)) for about 600K iterations and used the learned encoder and slot attention weights to initialize the corresponding modules of the VQ-SA model. We also experimented with end-to-end learning (Appendix A, Table 3 and Fig. 6). The model is trained in this mode, but converges more slowly, which complicates the setting of multiple experiments.

## 3.1 Set prediction

In the set prediction task, the model receives an image as an input and predicts target features as the unordered set of object vectors. The vectors of predicted and target features are matched using the Hungarian algorithm (Kuhn (1955)). To quantify the quality of the model, the Average Precision (AP) metric with a certain threshold $t$ is used. A detected object is considered being true positive if the set of its predicted properties exactly matches the ground truth object and the position of the detected object is predicted within a threshold $t$ relative to the ground truth object. The threshold $\infty$ ($\mathrm{AP}_\infty$) means we do not use the distance threshold. One of the difficult aspects of this task is the invariance of sets to permutations. This specific property is explicitly modeled in the architecture using the Slot Attention module (Locatello et al. (2020)), which turns a distributed representation of a whole scene into a set of object representations. The goal of our method is to model the distribution of object features. The number of vector spaces we use is equal to the implied number of generative factors. From each latent space, we sample one vector and concatenate them into the final object representation. As shown in Table 1, compared to the original Slot Attention based model, our approach allows the model to make more accurate predictions. It is also worth noting that the proposed model significantly improves the results for small thresholds, more than twice for the 0.25 threshold and more than two and a half times for the 0.125 threshold. Compared to a highly specialized model for the set prediction task, iDSPN (Zhang et al. (2022)), our proposed model VQ-SA demonstrates comparable results for the thresholds that are greater than one and moderate results for smaller thresholds. The reason is that the architecture of our model was not specially tuned to the set prediction task and could be used for other tasks, e.g., object discovery. Another key difference is that the learned discrete representations are disentangled (Section 4) and could be used to generate images with individual objects edited in an interpretable manner (Section 3.2).

**Setup**  To make correct comparisons with the Slot Attention model, we use the same hyperparameters during training: we use a batch size of 512, three iterations of Slot Attention, and 150,000

Table 1: Performance on the CLEVR object property set prediction task. For DSPN (Zhang et al. (2019)), Slot MLP and Slot Attention we use results from Locatello et al. (2020) (mean ± std for five seeds). The results of iDSPN are reported following Zhang et al. (2022). For the proposed model, mean and std are calculated for four seeds.

| Model | $AP_\infty$ (%) | $AP_1$ (%) | $AP_{0.5}$ (%) | $AP_{0.25}$ (%) | $AP_{0.125}$ (%) |
|---|---|---|---|---|---|
| Slot MLP | $19.8 \pm 1.6$ | $1.4 \pm 0.3$ | $0.3 \pm 0.2$ | $0.0 \pm 0.0$ | $0.0 \pm 0.0$ |
| DSPN T=30 | $85.2 \pm 4.8$ | $81.1 \pm 5.2$ | $47.4 \pm 17.6$ | $10.8 \pm 9.0$ | $0.6 \pm 0.7$ |
| DSPN T=10 | $72.8 \pm 2.3$ | $59.2 \pm 2.8$ | $39.0 \pm 4.4$ | $12.4 \pm 2.5$ | $1.3 \pm 0.4$ |
| Slot Attention | $94.3 \pm 1.1$ | $86.7 \pm 1.4$ | $56.0 \pm 3.6$ | $10.8 \pm 1.7$ | $0.9 \pm 0.2$ |
| VQ-SA (ours) | $\mathbf{96.1 \pm 0.4}$ | $\mathbf{91.2 \pm 0.5}$ | $\mathbf{71.8 \pm 2.3}$ | $\mathbf{22.2 \pm 2.1}$ | $\mathbf{2.4 \pm 0.2}$ |
| iDSPN | $98.8 \pm 0.5$ | $98.5 \pm 0.6$ | $98.2 \pm 0.6$ | $95.8 \pm 0.7$ | $76.9 \pm 2.5$ |

training iterations. The model is trained with the Adam (Kingma & Ba (2015)) optimizer with a learning rate of 0.0004. We also use learning rate warmup with an exponential decay schedule after that. The number of slots is equal to 10, as we use the CLEVR (Johnson et al. (2017)) dataset, except the images with more than 10 objects. The encoder architecture is shown in Appendix A, Table 5. The remaining model hyperparameters are presented in the Appendix A in Tables 4a and 4b.

## 3.2 OBJECT DISCOVERY

In the object discovery task, the model receives a raw image, separates distributed scene features into individual object features, and uses them to reconstruct objects, combining them into the original scene image. We use the Slot Attention module, which transforms input convolutional features into a set of vectors, and a convolutional decoder (Watters et al. (2019b)), which decodes each slot independently into the 4-channel image, where the first three channels are RGB color channels and the fourth channel is an unnormalized mask. To merge the reconstructed objects into the scene, the masks are normalized across all slots using Softmax and used as mixture weights. The original Slot Attention-based model achieves remarkable results in the object discovery task, while our extension allows the model not only to separate distributed features into object representations but also to separate distributed features of the objects themselves into representations of their properties.

Fig. 2 demonstrates the ability to manipulate individual objects in a scene with our model: each column corresponds to a change in one attribute (except the fourth column). Each next image in the column is obtained by changing only one discrete component of the object representation to the discrete latent variable from the same space.

**Setup**  We use the same training setup as the one in the original Slot Attention-based model: the main term in loss is the mean squared image reconstruction error, the batch size is equal to 64, and the optimizer and learning rate setup are the same as in Section 3.1. In this case, we utilize the CLEVR (Johnson et al. (2017)) dataset without images with more than six objects, so the number of slots is equal to 10. We also use an additional loss term, which aids disentanglement. It is described in Section 2.5. The encoder and decoder architectures are shown in Appendix A in Tables 5 and 6, respectively. The remaining model hyperparameters are presented in the Appendix A in Tables 4a and 4c. We also conducted experiments with the Tetrominoes Kabra et al. (2019) dataset for the object discovery task (Fig. 8 in Appendix A).

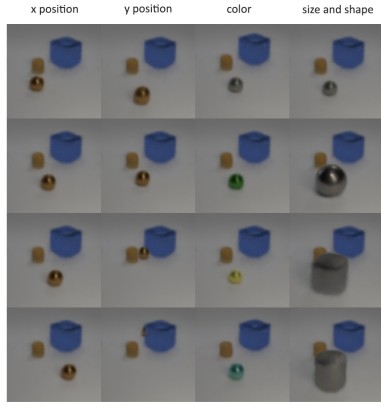

Figure 2: Learned discrete representations enable the manipulation of individual objects in the scene.

## 4 DISENTANGLED REPRESENTATIONS

To qualitatively evaluate the disentanglement of the trained discrete variables, we propose methods DQCF-micro and DQCF-macro (DQCF stays for Disentanglement Quality of Categorical generative Factors) that work when the value of generative factor is represented by the vector rather than a particular position in a vector. DQCF-micro evaluates disentanglement with respect to all other vectors from all discrete spaces, while DQCF-macro evaluates it on the level of discrete spaces. For DQCF-micro, we calculate for every set of objects from the validation data frequency of sampling each latent vector $e_j^k$ as the most similar vector. This statistics with Hungarian matching gives us frequency probabilities of each property $prop_p, p = 1 \dots P$ with values $value_v, v = 1 \dots V$ conditioned on the sampled latent vectors $p(prop_p = value_v | e_j^k)$. The example of this value for the first latent space $L_1$ is presented in Fig. 3. A strong closeness of the distribution of values of one property (e.g., values "small" and "large") to a uniform distribution means that this vector is not specific to objects with certain values of that property and does not contain information that is unique to that property.

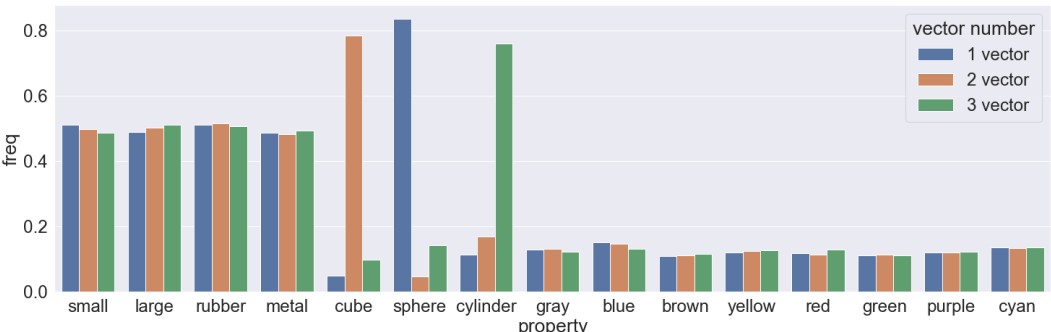

Figure 3: Results of DQCF-micro. Example of $p(prop_p = value_v | e_j^k)$ for embeddings from the first space. The probability is calculated as the frequency of objects with $value_v$ of property $prop_p$ for which the vector $e_j^k$ was sampled.

Therefore, for DQCF-macro, we further calculate the standard deviation over all values of corresponding properties and get the mean value over all vectors from the same space. The obtained values are presented in Fig. 4. Intuitively, these values show how much the change in the object properties affects the change in the distribution over a particular embeddings space. High values indicate that latent variables from this space contain information specific to this property, while low values indicate the opposite. It can be seen that the information specific to the property "color" is contained in the vectors of both the third and fourth spaces, i.e. they are entangled, while information about other properties is distributed over vectors of single spaces.

To deal with this entanglement, we use an additional KL-term in loss during training, described in Section 2.5. The difference in resulting disentanglement between models with and without this additional constraint is shown in 5.

There are three images: the original and two with a modified cylinder. Both modifications are obtained by changing one latent variable responsible for the size of the object, but in the case of the model trained without an additional KL-term, this led to a change in color, while in the other case, only the size changed.

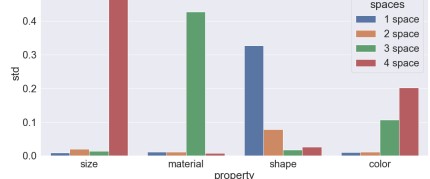

Figure 4: Results of DQCF-macro. Relation between variations in object properties and changes in the probability distribution over sets of latent variables.

## 5 ABLATION STUDIES

As ablation studies, we provide additional quantitative results on how constraints and extensions affect model performance in the supervised set prediction task.

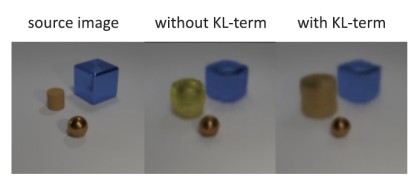

Figure 5: Example of the increasing object size with two models: one trained without using a KL-term and the other— using it.

We investigated how increasing the number of embeddings in each space affects the results. We did not get a noticeable change in the metrics by increasing the number of vectors in each space to eight (32 in total) and 16 (64 in total). Another modification was to use only a single common space that also does not affect the performance. The results for different settings are shown in Table 2.

Table 2: Performance comparison of the Slot Attention model (Locatello et al. (2020)), our proposed VQ-SA model, and its modifications with additional constraints and extensions. The number in parentheses, e.g., VQ-SA(32), indicates the total amount of embeddings we used in discrete latent spaces, i.e., eight embeddings in each for four latent spaces. "Cspace" means that we used only a single latent space with a given number of embeddings. Ablation studies' results are calculated for five seeds.

| Model | $AP_\infty$ (%) | $AP_1$ (%) | $AP_{0.5}$ (%) | $AP_{0.25}$ (%) | $AP_{0.125}$ (%) |
|---|---|---|---|---|---|
| Slot Attention | $94.3 \pm 1.1$ | $86.7 \pm 1.4$ | $56.0 \pm 3.6$ | $10.8 \pm 1.7$ | $0.9 \pm 0.2$ |
| VQ-SA (ours) | $\mathbf{96.1 \pm 0.4}$ | $91.2 \pm 0.5$ | $\mathbf{71.8 \pm 2.3}$ | $\mathbf{22.2 \pm 2.1}$ | $\mathbf{2.4 \pm 0.2}$ |
| VQ-SA(8, cspace) | $41.8 \pm 27.5$ | $38.2 \pm 26.5$ | $27.6 \pm 18.3$ | $7.0 \pm 4.4$ | $0.7 \pm 0.4$ |
| VQ-SA(16, cspace) | $83.3 \pm 7.7$ | $79.3 \pm 7.2$ | $58.8 \pm 4.5$ | $15.3 \pm 1.2$ | $1.5 \pm 0.1$ |
| VQ-SA(32, cspace) | $92.3 \pm 1.4$ | $88.1 \pm 1.4$ | $66.0 \pm 2.4$ | $17.7 \pm 1.6$ | $1.8 \pm 0.2$ |
| VQ-SA(64, cspace) | $94.2 \pm 0.6$ | $90.3 \pm 0.5$ | $69.2 \pm 2.6$ | $19.1 \pm 2.2$ | $1.9 \pm 0.3$ |
| VQ-SA(32) | $95.9 \pm 0.2$ | $\mathbf{92.1 \pm 0.4}$ | $70.4 \pm 1.6$ | $18.9 \pm 1.4$ | $1.9 \pm 0.2$ |
| VQ-SA(64) | $96.1 \pm 0.1$ | $92.1 \pm 0.3$ | $69.6 \pm 1.1$ | $18.5 \pm 1.0$ | $1.8 \pm 0.1$ |

## 6 RELATED WORK

**Disentanglement**    There is no conventional definition of a disentangled representation. The intuitive idea behind the term "disentanglement" is the one about representations that capture and separate generative factors in the data (Bengio et al. (2013); Higgins et al. (2018)). Generative factors of the dataset are a set of independent factors that can describe any sample from the dataset. Metrics of disentanglement such as Beta-VAE (Higgins et al. (2017b)), FactorVAE (Kim & Mnih (2018)), DCI (Eastwood & Williams (2018b)), and SAP (Kumar et al. (2018b)) were designed to reflect the definition of disentanglement representations from Bengio et al. (2013).

Most of the recent approaches to obtaining disentangled representations are based on the Variational Autoencoder framework, which consists of an encoder, that maps the data samples into their latent representations, and a decoder that maps given representations into source data space. VAE tends to match the distribution of the latent representation of the input data and the standard Gaussian distribution. Thus, each representation is generated from a continuous distribution and may not reflect the discrete nature of some generative factors. Beta-VAE (Higgins et al. (2017b); Burgess et al. (2018)) and Factor-VAE (Kim & Mnih (2018)) as modifications of original VAE use additional constraints during training to enforce better disentanglement. As an architecturally different approach, InfoGAN (Chen et al. (2016)) maximizes mutual information between a subset of latent variables and the generated samples to aid disentanglement.

**Vector quantized representations**    Using discrete latent representation can be seen as a way to gain some advantages in modeling discrete generative factors of data. Assuming that the dimensionality of the discrete factors is known, it is possible to enhance disentanglement by inducing semantic factor biases from known factors (Locatello et al. (2019)). Vector Quantized Variational Autoencoder (VQ-VAE) (van den Oord et al. (2017)) is an approach to learn discrete latent variables by splitting encoded inputs into fixed-size embeddings and assigning them to the nearest vectors from a learnable codebook. The prior distribution of discrete representations is modeled by a separate autoregressive model. VQ-GAN (Esser et al. (2020)) takes the idea of VQ-VAE and extends it through the use of transformers, discriminator and perceptual loss, achieving remarkable image generation results. One of the crucial challenges of Vector Quantized models is achieving high codebook usage, since typically, there is a noticable part of vectors that are rarely used. Yu et al. (2021) claims that mapping all latent variables on a sphere by normalization and using lower-dimensional lookup space can significantly improve the codebook usage with training stability and reconstruction quality. Shin et al.

(2021) investigates how forcing latent variables to be orthogonal brings translation equivariance in the quantized space with increased performance in image-to-text and text-to-image generation tasks. In contrast to the improvements described in Yu et al. (2021), this technique reduces the number of latent vectors used.

**Set prediction** Neural networks for sets are applied in various fields: object detection (Carion et al. (2020)), point cloud generation (Achlioptas et al. (2017)), molecule generation (Simonovsky & Komodakis (2018)), and speaker diarization (Fujita et al. (2019)). Although the set structure is natural for use in many data, traditional deep learning models are not inherently suitable for representing sets. There are some approaches that are built to reflect the unordered nature of sets: the Deep Set Prediction Network (DSPN) (Zhang et al. (2019)) reflects permutation symmetry by running an inner gradient descent loop that changes a set to encode more similarly to the input; iDSPN Zhang et al. (2022) is an improved version of DSPN with approximate implicit differentiation that provides better optimizations with faster convergence and state-of-the-art performance on the CLEVR dataset. Slot Attention (Locatello et al. (2020)) and TSPN (Kosiorek et al. (2020)) use set-equivariant self-attention layers to represent the structure of sets.

**Object discovery** The discovery of objects in a scene in an unsupervised manner is a crucial aspect of representation learning and a desirable part of a binding problem solution. Works as MONET (Burgess et al. (2019)), IODINE (Greff et al. (2019)), and GENESIS (Engelcke et al. (2020)) are built upon the Variational Autoencoder (VAE) framework (Kingma & Welling (2014); Rezende et al. (2014)). MONET uses the attention network that generates masks and conditions VAE on these masks. IODINE models an image as a spatial Gaussian mixture model to jointly infer the object representation and segmentation. Compared to MONET and IODINE, GENESIS explicitly models dependencies between scene components that allow the sampling of novel scenes. MONET, IODINE, and GENESIS use multiple steps to encode and decode an image, while Slot Attention (and its sequential extension for video (Kipf et al. (2021)) uses one step but performs an iterative procedure inside this step. The useful property of Slot Attention is that it produces the set of output vectors (slots) with permutation symmetry. Slots group input information and could be used in unsupervised tasks (object discovery) and supervised tasks (set prediction). GENESIS-v2 (Engelcke et al. (2021)), a development of the GENESIS model, uses attention masks similarly to Locatello et al. (2020). Another approach utilizes Generative Adversarial Networks (GAN) (Goodfellow et al. (2014)) by structuring the generative process (van Steenkiste et al. (2018)). That enables learning about individual objects without supervision. Our work is closely related to Slot Attention, but we move further and improve slot representation by quantization.

## 7 CONCLUSION AND DISCUSSION

In this paper, we propose the VQ-SA model that utilizes the idea of slot object representation and models non-overlapping sets of low-dimensional discrete variables, sampling one vector from each to obtain the latent representation of the object. Such representation allows one not only to separate scene distributed features into object representations but also to separate distributed features of the objects themselves into representations of their properties. Our model achieves state-of-the-art results among the class of object-centric methods on a set prediction task. We also demonstrate that through manipulation with learned discrete representations, we can generate scenes with objects with the given property. To qualitatively evaluate the disentanglement in the object discovery task we propose DQCF-micro and DQCF-macro methods.

The important feature of our model is that we use the number of latent discrete spaces equal to the number of generative factors in the data. Thus, our model cannot be applied out of the box to data with a different number of generative factors. As with most object-oriented models, we show results on relatively simple data. Modifying our model for more complex scenes both from real life and from simulators, e.g., AI2Thor (Kolve et al. (2017)), is an interesting and challenging task and will be considered in future works.

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

## A APPENDIX

Table 3: Performance comparison of the Slot Attention model (Locatello et al. (2020)), our proposed VQ-SA model and VQ-SA model trained for 150K steps with (VQ-SA-p 150K) and without (VQ-SA 150K) pretrained Slot Attention Module in the supervised set prediction task.

| Model | $AP_\infty$ (%) | $AP_1$ (%) | $AP_{0.5}$ (%) | $AP_{0.25}$ (%) | $AP_{0.125}$ (%) |
|---|---|---|---|---|---|
| Slot Attention | $94.3 \pm 1.1$ | $86.7 \pm 1.4$ | $56.0 \pm 3.6$ | $10.8 \pm 1.7$ | $0.9 \pm 0.2$ |
| VQ-SA (ours) | $\mathbf{96.1 \pm 0.4}$ | $91.2 \pm 0.5$ | $71.8 \pm 2.3$ | $22.2 \pm 2.1$ | $2.4 \pm 0.2$ |
| VQ-SA 150K | 87.0 | 82.2 | 63.3 | 19.7 | 2.1 |
| VQ-SA-p 150K | 96.0 | **91.7** | **78.3** | **30.4** | **3.8** |

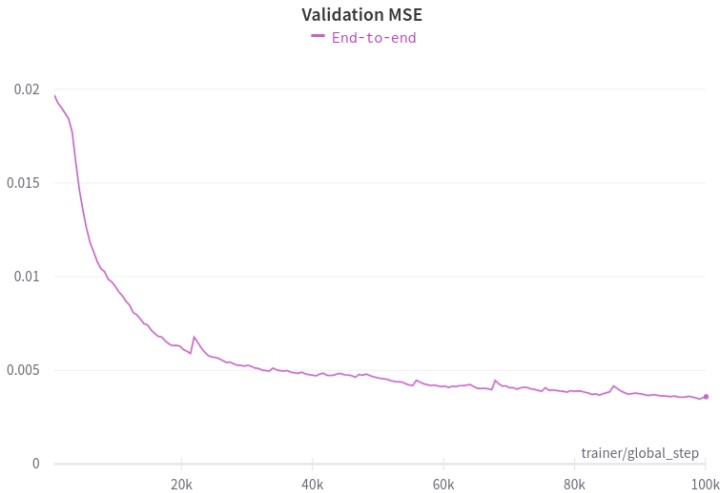

Figure 6: Learning curve for the object discovery task model with end-to-end learning.

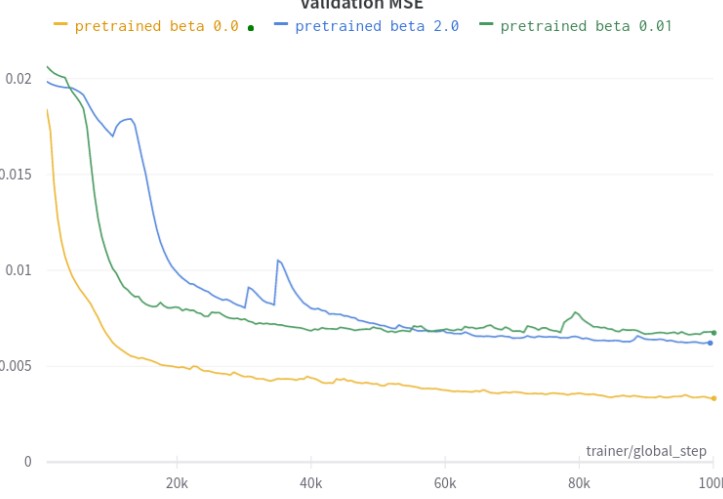

Figure 7: Learning curves for different $\beta$ values for model in the object discovery task with pretrained slot attention weights initialization.

Table 4: Hyperparameters of the VQ-SA model for the set prediction and the object discovery tasks.

(a) Shared hyperparameters.

| Name | Value |
|------|-------|
| AdamW: $\beta_1$ | 0.9 |
| AdamW: $\beta_2$ | 0.999 |
| AdamW: $\epsilon$ | 1e-08 |
| AdamW: learning | 0.999 |
| AdamW: weight decay | 0.01 |
| OneCycleLR: pct. start | 0.05 |
| Slot dim. | 64 |

(b) Hyperparameters for the set prediction task.

| Name | Value |
|------|-------|
| Batch size | 512 |
| Train steps | 150K |

(c) Hyperparameters for the object discovery task.

| Name | Value |
|------|-------|
| Batch size | 64 |
| Train steps | 500K |

Table 5: Architecture of the CNN encoder for the experiments on CLEVR dataset for set property prediction and object discovery tasks. Set prediction uses stride of 2 in the layers with *, while object discovery model uses stride of 1 in these layers.

| Layer | Channels | Activation | Params |
|-------|----------|------------|--------|
| Conv2D $5 \times 5$ | 64 | ReLU | stride: 1 |
| Conv2D $5 \times 5$ | 64 | ReLU | stride: 1/2* |
| Conv2D $5 \times 5$ | 64 | ReLU | stride: 1/2* |
| Conv2D $5 \times 5$ | 64 | ReLU | stride: 1 |
| Position Embedding | - | - | absolute |
| Flatten | - | - | dims: w, h |
| LayerNorm | - | - | - |
| Linear | 64 | ReLU | - |
| Linear | 64 | - | - |

Table 6: Spatial broadcast decoder for object discovery task for CLEVR dataset

| Layer | Channels/Size | Activation | Params |
|-------|---------------|------------|--------|
| Spatial Broadcast | $8 \times 8$ | - | - |
| Position Embedding | - | - | absolute |
| ConvTranspose2D $5 \times 5$ | 64 | ReLU | stride: 2 |
| ConvTranspose2D $5 \times 5$ | 64 | ReLU | stride: 2 |
| ConvTranspose2D $5 \times 5$ | 64 | ReLU | stride: 2 |
| ConvTranspose2D $5 \times 5$ | 64 | ReLU | stride: 2 |
| ConvTranspose2D $5 \times 5$ | 64 | ReLU | stride: 1 |
| ConvTranspose2D $3 \times 3$ | 4 | - | stride: 1 |
| Split Channels | RGB (3), mask (1) | Softmax on masks (slots dim) | - |
| Combine components | - | - | - |

Table 7: Spatial broadcast decoder for object discovery task for Tetrominoes dataset

| Layer | Channels/Size | Activation | Params |
|-------|---------------|------------|--------|
| Spatial Broadcast | $35 \times 35$ | - | - |
| Position Embedding | - | - | absolute |
| ConvTranspose2D $3 \times 3$ | 32 | ReLU | stride: 1 |
| ConvTranspose2D $3 \times 3$ | 32 | ReLU | stride: 1 |
| ConvTranspose2D $3 \times 3$ | 32 | ReLU | stride: 1 |
| ConvTranspose2D $3 \times 3$ | 4 | - | stride: 1 |
| Split Channels | RGB (3), mask (1) | Softmax on masks (slots dim) | - |
| Combine components | - | - | - |

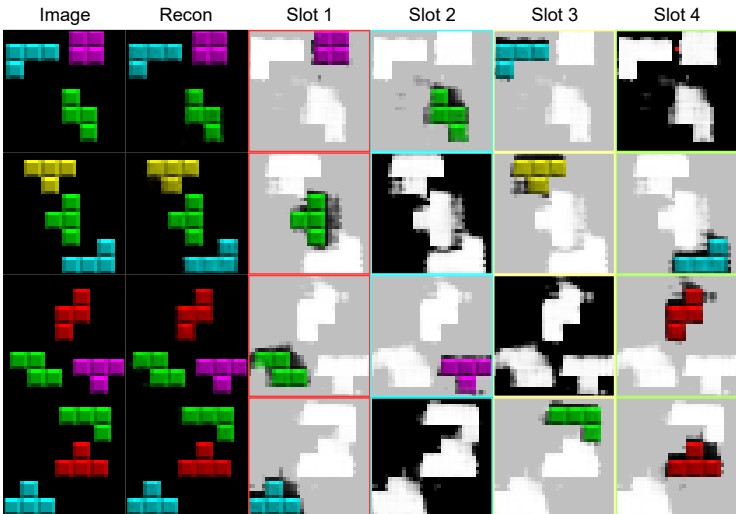

Figure 8: Visualization of per-slot reconstructions and alpha masks in the object discovery training setting for Tetrominoes dataset.

