# OpenReview forum: "Quantized Disentangled Representations for Object-Centric Visual Tasks"
_ICLR.cc/2023/Conference — Submitted to ICLR 2023_

### Official Review · Reviewer_88V8 · 2022-10-24

**Confidence:** 4
**Correctness:** 2
**Technical Novelty And Significance:** 2
**Empirical Novelty And Significance:** 2
**Recommendation:** 3

**Clarity, Quality, Novelty And Reproducibility:**

The paper lacks clarity in several key illustrations of designs (e.g. definition of DFCQ) and might cause problems in understanding. The idea of quantizing slot representations is new however the current experimental results show the limited significance of the learned VQ-SA as a model for better disentanglement in object-centric learning. Codes are provided, however, with limited description on implementation details in the text.

**Strength And Weaknesses:**

[+] The idea of vector quantizing slot representations for better disentangling representations is new.

[+] The resulting model does show better performance compared with the vanilla slot attention model.

[-] The major concern of this paper lies in the justification of claims in this paper and experiments. The authors are motivated to perform vector quantization to better learn disentangled object-centric representations. However, they only tested the resulting model on the CLEVR dataset on the set-prediction task which shows the limited significance of the design. The disentangling mechanism (KL) is also not quantitatively evaluated in ablative studies. This makes the overall claims of the proposed VQ-SA not fully addressed and justified.

[-] The authors might want to elaborate more on disentangled representations as it is a critical factor in this paper and show their significance. The current sec.4 does not fully show the uniqueness of learning discrete spaces and does not make a direct comparison of disentanglement with ones that do not leverage a quantization module. The description of DQCF-micro and DQCF-macro is also a bit hard to follow in the text, especially given notations are not properly defined before the illustrations.

**Summary Of The Paper:**

This paper proposes to use a latent space to quantize object-centric representations learned with slot attention for better disentanglement. The authors adopted a similar idea of VQ-VAE and initialized learnable codebooks for each slot representation generated from slot attention to obtain the vector-quantized representation of slot representations.  With experiments, the authors show that their methods can outperform slot attention on set prediction task in CLEVR.

**Summary Of The Review:**

Given that the current experimental results can not fully justify the claims of the authors, I'm recommending rejection and suggest the authors design better tasks/settings for illustrating the effectiveness of the proposed VQ-SA, not only from a qualitative perspective on the analysis of latent spaces. The authors might also want to step further from CLEVR to more complex datasets (both real and synthetic) for evaluating the learned representations.

---

> ### Author Response · Authors · 2022-11-19
> **Response to Reviewer 88V8**
>
> Thank you for your comment. It is very helpful for us to continue to improve this work.
>
> **W:** The major concern of this paper lies in the justification of claims in this paper and experiments. The authors are motivated to perform vector quantization to better learn disentangled object-centric representations. However, they only tested the resulting model on the CLEVR dataset on the set-prediction task which shows the limited significance of the design. The disentangling mechanism (KL) is also not quantitatively evaluated in ablative studies. This makes the overall claims of the proposed VQ-SA not fully addressed and justified.
>
> - **Re:** We additionally conducted experiments with the Tetrominos dataset for the object discovery task. Examples of image reconstructions and predicted masks are shown in Appendix A, Figure 8.
>
> **W:** The authors might want to elaborate more on disentangled representations as it is a critical factor in this paper and show their significance. The current sec.4 does not fully show the uniqueness of learning discrete spaces and does not make a direct comparison of disentanglement with ones that do not leverage a quantization module. The description of DQCF-micro and DQCF-macro is also a bit hard to follow in the text, especially given notations are not properly defined before the illustrations.
>
> - **Re:** We have changed the description of the proposed qualitative evaluation methods, which we hope will make it easier to understand. For DQCF-micro, we calculate for every set of objects from the validation data frequency of sampling each latent vector $e^{k}_{j}$ as the most similar vector.  This statistics with Hungarian matching gives us frequency probabilities of each property $prop_p, p=1 \dots P$ with values $value_v, v = 1\dots V$ conditioned on the sampled latent vectors $p(prop_p=value_v | e^k_j)$.
>
>
> We also have added model training hyperparameters to Appendix A in Table 4 the description of architectures in Tables 5 and 6.

---

> > ### Comment · Reviewer_88V8 · 2022-12-01
> > **Post-rebuttal response**
> >
> > Thank the authors for the rebuttal and clarifications. However, I still think that the current experiments are not sufficient for justifying the authors' claims. The authors could try more challenging datasets (e.g. by manipulating existing datasets to add the features you wanted) to verify the claims in future revisions.

---

### Official Review · Reviewer_1Nh6 · 2022-10-24

**Confidence:** 5
**Correctness:** 2
**Technical Novelty And Significance:** 2
**Empirical Novelty And Significance:** 1
**Recommendation:** 1

**Clarity, Quality, Novelty And Reproducibility:**

The writing reads like an early draft. Please see the following comments and suggestions:

1. Inline citations are weirdly done. Section 6, “Object discovery” is impossible to read because of recurring citations. You don’t need to cite a model more than once in the same in the same paragraph.
2. Section 2.3:
    - Why not denote each latent space as $L_k$ if you’re using K to denote the number of latent spaces?
    - You don’t seem to set the number of embeddings $n_{L_i}$ differently across the latent spaces. So why not drop the ${L_i}$ subscript and simply use $n$? This would also be consistent with your equations.
    - Matrix multiplication is not denoted using $\times$. It would be sufficient to say $S’ = MS$. Likewise for $sim^i$.
    - It is unusual to write $S^D = [e^i, …, e^K]$. Generally this would be written $S^D = [e^1, …, e^K]$.
3. Section 2.4 typo: “MPL” -> “MLP”?
4. Section 2.5: you introduce q(L_i | S’) for the first time here. Is it just a categorical distribution over n embeddings parameterized by ${y^i_1, …, y^i_n}$?
5. Section 5 typo: “qualitative” -> “quantitative”?
6. Please make sure all equations have a number.

**Strength And Weaknesses:**

Pros:
- The set prediction results show a marginal improvement over Slot Attention.
- Code is available in the supplementary materials.

Cons:
- The results are pretty basic on CLEVR alone. Only a single example image is used throughout the paper (for the object discovery and disentanglement results). There are no additional images in the Supplementary material.
- The proposed evaluation for disentanglement (DQCF-micro and DQCF-macro) is purely qualitative and involves inspecting histograms (for DQCF-mico, this could be over several latent spaces). Surely deviation from uniform could be quantified as a metric?
- The paper is not clear on several technical aspects (see questions below). It reads like an early draft at the moment.

**Summary Of The Paper:**

This work proposes a version of Slot Attention with vector-quantized representations, focussing on object- as well as feature-level disentanglement. The paper also proposes a pair of techniques (DQCF-micro and DQCF-macro) to look at disentanglement when generative factors are encoded as vectors.

**Summary Of The Review:**

The paper is certainly not ready for publication. The results need fleshing out and the writing needs a few more iterations.

Here are some questions to help the paper in future iterations:
1. __End-to-end__? Figure 1 shows the “three stages” of the VQ-SA pipeline. Could you confirm whether the stages are run one after the other (freezing weights as you go), or if the pipeline is run end to end?
2. __Prior work__: This isn’t the first work that uses discrete representations with Slot Attention. How does this work relate to Singh et al. 2021 where they also use a discrete VAE?
3. __Traversals__: Figure 2: what exactly do you mean by manipulating a particular attribute in your model? Since each slot representation $S^D$ is a weighted sum of embeddings (concatenated across latent spaces), I don’t understand how you can manipulate the weighted sum. Do you replace the weights $y^i$ with a one-hot lookup?
4. __KL term__: I assume Figure 4 corresponds to the model trained “without KL-term” (shown in Figure 5). Could you also share DQCF-macro results for the model which achieves better disentanglement (“with KL-term”)? What is the effect of boosting the weight of the KL loss?
5. __Multiple latent spaces__: In Table 2, it appears that using a single latent space with 32 embeddings yields better set prediction performance than 4 different latent spaces. Why is that? Is disentanglement the only rationale of using multiple latent spaces?

---

> ### Author Response · Authors · 2022-11-19
> **Response to Reviewer 1Nh6**
>
> Thank you for the detailed review and insightful comments.
>
> **W:** The results are pretty basic on CLEVR alone. Only a single example image is used throughout the paper (for the object discovery and disentanglement results). There are no additional images in the Supplementary material.
>
> - **Re:** We additionally conducted experiments with the Tetrominos dataset for the object discovery task. Examples of image reconstructions and predicted masks are shown in Appendix A, Figure 8.
>
> **W:** The proposed evaluation for disentanglement (DQCF-micro and DQCF-macro) is purely qualitative and involves inspecting histograms (for DQCF-mico, this could be over several latent spaces). Surely deviation from uniform could be quantified as a metric?
>
> - **Re:** We do not refer to DQCF-micro and DQCF-macro (DQCF stays for Disentanglement Quality of Categorical generative Factors) as disentanglement metrics, but they can evaluate qualitatively disentanglement of representations.
>
> **W:** The paper is not clear on several technical aspects. It reads like an early draft at the moment.
>
> - **Re:** We have rewritten Section 2.3, introduced new index notations and added definitions in appropriate places, which we hope will improve the readability of the text. We have also given the rest of the notation in the text in accordance with Section 2.3 and numbered the formulas. We fixed typos and removed duplicate links from Section 6.
>
> **Clarity, Quality, Novelty And Reproducibility:**
>
> **End-to-end?** Figure 1 shows the “three stages” of the VQ-SA pipeline. Could you confirm whether the stages are run one after the other (freezing weights as you go), or if the pipeline is run end to end?
>
> - **Re:** We first train the Slot Attention model for a specific task (Set Prediction or Object Discovery). Then we used pre-trained encoder and Slot Attention modules, add Slot Quantization and task-depended head and fine-tuned the model. We do not freeze already trained parts. We also experimented with end-to-end training, the model converging but over more steps. The results for end-to-end training are presented in Appendix A, Table 3. To improve the explanation of the proposed architecture in Figure 1, we have presented models for set prediction and object detection tasks separately.
>
> **Prior work:** This isn’t the first work that uses discrete representations with Slot Attention. How does this work relate to Singh et al. 2021 where they also use a discrete VAE?
>
> - **Re:** Singh et al. 2021 uses dVAE to discretize the whole distributed feature map from the CNN encoder before a slots assignment. Our approach involves discretization of slots.
>
> **Traversals:** Figure 2: what exactly do you mean by manipulating a particular attribute in your model? Since each slot representation  is a weighted sum of embeddings (concatenated across latent spaces), I don’t understand how you can manipulate the weighted sum. Do you replace the weights  with a one-hot lookup?
>
> - **Re:** We assume choosing one latent vector from one latent space (since we decrease Gumbel-Softmax sampling temperature going to the singular distributions).
>
> **Multiple latent spaces:** In Table 2, it appears that using a single latent space with 32 embeddings yields better set prediction performance than 4 different latent spaces. Why is that? Is disentanglement the only rationale of using multiple latent spaces?
>
> - **Re:** We supplemented the ablation studies in Table 2. In these experiments. We used different configurations of latent space: without division into several spaces (cspace) and with division into the number of spaces equal to the number of categorical generative factors. An increase in the number of vectors in spaces does not lead to a significant increase in metrics.

---

### Official Review · Reviewer_Eih1 · 2022-10-25

**Confidence:** 4
**Correctness:** 3
**Technical Novelty And Significance:** 2
**Empirical Novelty And Significance:** 2
**Recommendation:** 3

**Clarity, Quality, Novelty And Reproducibility:**

* The paper was not extremely clear, and several sections were quite hard to follow.
* As explained above, I do not think the results and the way the model is presented is of the standard expected by ICLR in this current draft.
* I have not seen a model combining SlotAttention with VQ-VAE yet, so the work presented appears novel to me AFAIK.

**Strength And Weaknesses:**

1. The presentation of the model as 2 independent tasks and entirely different methods is not very helpful. I would have expected the model to be trained on image reconstruction only, and demonstrate that one can make use of the discrete representations to perform the set prediction post-hoc.
   1. Figure 1 shows the setup as one single model diagram, even though they are done entirely separately and in different training stages.
   2. Section 1 and 2 are not very clear in presenting this choice.
   3. The Set Prediction task is not presented well enough. What are these thresholds?
   4. Overall I found the paper hard to follow, even though the idea is rather simple.
2. Combining VQ-VAE with SlotAttention is a good idea, and I think some of the choices they made in how to do so makes sense, however this should be done end to end and with less assumptions to be really impactful.
   1. What happens when you learn everything together? In particular if this was made to work when trying to do reconstruction, this would be a valuable piece of research.
   2. The number of categorical distributions being fixed to the number of generative factors, and the number of categories to be equal to the number of values per factor is too much supervision.
      1. What happens if you use more?
   3. The fact that continuous variables are handled independently and entirely differently from categorical ones is too much supervision.
      1. What happens if you just use VQ for everything? Obviously you would need to use many for this to make sense.
3. The proposed disentanglement metrics were confusing and I would have assumed that computing the discrete mutual information would have directly done the same?
   1. Can you comment on how these differ?
   2. Figure 3 and 4 were not very clear to me, and feel like they belong to the Appendix? A table could replace Figure 4 and be more informative.
4. Nits:
   1. The math in section 2.3 uses cross products instead of dot products. This should be changed.
   2. I could not see an Appendix with details of the architecture and training setup. There are not enough details about the model in the main text to reproduce this work.


**Summary Of The Paper:**

This paper combines Slot Attention with VQ-VAE (although in a stepwise training fashion) and shows results on CLEVR and on set prediction tasks.


**Summary Of The Review:**

Overall, I think this paper tries to do something interesting, but it makes several arbitrary and limiting choices, which seriously hinder the usefulness of the model, for example the decision to train SlotAttention first, then the VQs second. It also assumes quite a lot of knowledge about the task in various places, which reduces its use as a real unsupervised method. Finally, the current presentation is not as clear as it could be and I found it hard to follow.

Hence in this current form I do not believe this work reaches the standard expected by ICLR.

---

> ### Author Response · Authors · 2022-11-19
> **Response to Reviewer Eih1**
>
> We thank the reviewer for carefully reviewing our manuscript.
>
> **W:** The presentation of the model as 2 independent tasks and entirely different methods is not very helpful. I would have expected the model to be trained on image reconstruction only, and demonstrate that one can make use of the discrete representations to perform the set prediction post-hoc.
>
> - **Re:** The presentation of the model in the form of two tasks is caused by the fact that we followed the same style of presentation as in the original Slot Attention. In the initial model, a completely different set of parameters is trained for each task. In our architecture, only the task-specific head of the model changes for a new task.
>
> **Figure 1 shows the setup as one single model diagram, even though they are done entirely separately and in different training stages.**
> - **Re:** In Figure 1, we have presented models for set prediction and object detection tasks separately, which we hope will improve the explanation of the proposed architecture.
>
> **The Set Prediction task is not presented well enough. What are these thresholds?**
> - **Re:** We have refined the description of the set prediction task. To quantify the quality of the model, we use the Average Precision (AP) metric with a certain threshold $t$. A detected object is considered being true positive if the set of its predicted properties exactly matches the ground truth object and the position of the detected object is predicted within a threshold $t$ relative to the ground truth object. The threshold $\infty$ (AP$_{\infty}$) means we do not use the distance threshold.
>
> **W:** Combining VQ-VAE with SlotAttention is a good idea, and I think some of the choices they made in how to do so makes sense, however this should be done end to end and with less assumptions to be really impactful.
>
> **What happens when you learn everything together? In particular if this was made to work when trying to do reconstruction, this would be a valuable piece of research.**
> - **Re:** We experimented with end-to-end training, the model converging but over more steps. The results  for end-to-end training are presented in Appendix A, Table 3.
>
> **The number of categorical distributions being fixed to the number of generative factors, and the number of categories to be equal to the number of values per factor is too much supervision. What happens if you use more?**
> - **Re:** We supplemented the ablation studies in Table 2. In these experiments. We used different configurations of latent space: without division into several spaces (cspace) and with division into the number of spaces equal to the number of categorical generative factors. An increase in the number of vectors in spaces does not lead to a significant increase in metrics.
>
> **The fact that continuous variables are handled independently and entirely differently from categorical ones is too much supervision.
> What happens if you just use VQ for everything? Obviously you would need to use many for this to make sense.**
> - **Re:** In our experiments, the use of only categorical spaces led to the fact that the model did not converge.
>
> **W:** The proposed disentanglement metrics were confusing and I would have assumed that computing the discrete mutual information would have directly done the same?
>
> **Can you comment on how these differ?**
>
> - **Re:** Mutual information gives us information about how much one discrete latent variable tells us about another. Our measure of disentanglement tells how one discrete latent variable connected with some generative factor. We do not refer to DQCF-micro and DQCF-macro (DQCF stays for Disentanglement Quality of Categorical generative Factors) as disentanglement metrics, but they can evaluate qualitatively disentanglement of representations.
>
> **Figure 3 and 4 were not very clear to me, and feel like they belong to the Appendix? A table could replace Figure 4 and be more informative.**
>
> - **Re:** As mentioned above, DQCF-micro and DQCF-macro are not metrics of disentanglement. They can only serve as a qualitative measure of disentanglement, so we assume that a histogram visualization is more appropriate than a table view.
>
>
> **The math in section 2.3 uses cross products instead of dot products. This should be changed.**
>
> - **Re:** We have corrected these designations.
>
> **I could not see an Appendix with details of the architecture and training setup. There are not enough details about the model in the main text to reproduce this work.**
>
> - **Re:** We have added model training parameters to Appendix A in Table 4 and the description of architectures in Tables 5 and 6.

---

### Official Review · Reviewer_vXo7 · 2022-10-29

**Confidence:** 5
**Clarity, Quality, Novelty And Reproducibility:** Please see above.
**Correctness:** 3
**Technical Novelty And Significance:** 2
**Empirical Novelty And Significance:** 2
**Recommendation:** 3

**Strength And Weaknesses:**

Strengths:
- The proposed work seems to be a reasonable extension of the slot-attention model, and the empirical results on set prediction show some extent of improvements over the baseline.

Weaknesses:
- The writing is generally not as clear as it could be. The most important part of the paper should be Sec. 2.3, however, it's unnecessarily hard to follow, and Figure 1 is not very helpful here. The notations of introduced variables are not introduced in a more natural way, for example, the $e_j^i$ term is not described in a clear way in its first appearance.
- In Sec 2.5, the authors propose to use beta-VAE style loss to encourage the latent space to be disentangled. However, there's no mention and discussion on the weighting effect, i.e. $\beta>1$, which plays a crucial role.
- In object discovery tasks, only qualitative results are provided, and no quantitative evaluation is included, which is not convincing enough, especially when only CLEVR dataset is considered.

**Summary Of The Paper:**

This work proposed to combine the idea of slot-attention and vector quantization to learn discrete object-centric representation of visual scenes. The proposed model utilizes slot-attention to decompose the image into a set of object-centric slots, and then transform each inferred slot into a few concatenated vectors by vector quantization from a learned codebook. Both set prediction and object discovery tasks are evaluated to show the effectiveness of the proposed model, while particular efforts are made to show the ability to discover disentangled subspace in the latent space partitioned by the discrete quantization.

**Summary Of The Review:**

This work proposes an approach for learning discrete object-centric representation by combining existing ideas, I believe more thorough evaluation and better clarity are needed in this reviewing cycle.

---

> ### Author Response · Authors · 2022-11-19
> **Response to Reviewer vXo7**
>
> We thank the reviewer for carefully reviewing our manuscript.
>
> **W:** The writing is generally not as clear as it could be. The most important part of the paper should be Sec. 2.3, however, it's unnecessarily hard to follow, and Figure 1 is not very helpful here. The notations of introduced variables are not introduced in a more natural way, for example, the $e_j^i$ term is not described in a clear way in its first appearance.
>
> - **Re:** We have rewritten Section 2.3, introduced new index notations and added definitions in appropriate places, which we hope will improve the readability of the text. We have also given the rest of the notation in the text in accordance with Section 2.3. In Figure 1, we have presented models for set prediction and object detection tasks separately, which we hope will improve the explanation of the proposed architecture.
>
> **W:** In Sec 2.5, the authors propose to use beta-VAE style loss to encourage the latent space to be disentangled. However, there's no mention and discussion on the weighting effect, i.e. $\beta  > 1$, which plays a crucial role.
>
> - **Re:** We have added the results of ablation experiments with different $\beta$ values in Appendix A, Figure 6.

---

### Author Response · Authors · 2022-11-19
**General Response**

Thank you to all reviewers for taking the time and effort to read through our work and giving comments and feedback.

We posted a new version where we took considered all the main remarks and we ask everyone to briefly familiarize themselves with it. All main changes are highlighted in color for convenience.

We look forward to engaging the reviewers and are committed to address any further concerns.

---

### Decision · Program_Chairs · 2023-01-20

**Decision:**

Reject

**Justification For Why Not Higher Score:**

The authors have improved the clarity of the paper in their revision, but the main concerns around experimental validation remain. Overall, even though the idea is interesting and warrants further exploration, the paper in its current form is not ready for publication.

**Justification For Why Not Lower Score:**

N/A

**Metareview: Summary, Strengths And Weaknesses:**

This paper introduces Vector Quantized Slot Attention (VQ-SA), a method that applies vector quantization on the learned slots of Slot Attention [1]. VQ-SA is empirically evaluated on supervised set prediction and unsupervised object discovery on synthetic multi-object datasets such as CLEVR.

The idea of combining vector quantization with Slot Attention to address shortcomings of the module is interesting and the authors show some quantitative as well as qualitative improvements, but as all the reviewers point out, the paper in its current form has significant weaknesses: the claims of the paper are not sufficiently experimentally verified (only on very simple synthetic datasets with, for some claims, solely qualitative results), several technical aspects are unclear, and the writing/presentation could be improved.

Given the above limitations, which still hold after the author rebuttal, I recommend rejecting this paper in its current form.

[1] Locatello et al., Object-Centric Learning with Slot Attention (NeurIPS 2020)